# Mental health and gender-based violence: An exploration of depression, PTSD, and anxiety among adolescents in Kenyan informal settlements participating in an empowerment intervention

**Rina Friedberg**[1]*, **Michael Baiocchi**[2,3¤a], **Evan Rosenman**[4¤b], **Mary Amuyunzu-Nyamongo**[5¤c], **Gavin Nyairo**[5¤c], **Clea Sarnquist**[6¤d]

1 LinkedIn, Data Science and Applied Research (all work completed while at Department of Statistics, Stanford University), Stanford, CA, United States of America, 2 Stanford Prevention Research Center, Stanford, CA, United States of America, 3 Department of Statistics, Stanford University, Stanford, CA, United States of America, 4 Harvard Data Science Initiative, Cambridge, MA, United States of America, 5 African Institute for Health and Development, Nairobi, Kenya, 6 Department of Pediatrics, Stanford University School of Medicine, Stanford, CA, United States of America

¤a Current address: 1265 Welch Rd, Palo Alto, California, United States of America
¤b Current address: 8 Story Street Suite 380, Cambridge, Massachusetts, United States of America
¤c Current address: 7th Floor Suite B, Commodore, Wood Avenue/Kindaruma Road Junction, Box 45259, Nairobi, Kenya
¤d Current address: 300 Pasteur Drive, Stanford, California, United States of America
* rinafriedberg@gmail.com

**Data Availability Statement:** These data are highly sensitive, due to (a) sexual assault reporting, (b)

## Abstract

### Objective

This study examines the prevalence of depression, anxiety, and post-traumatic stress disorder (PTSD) among adolescents attending schools in several informal settlements of Nairobi, Kenya. Primary aims were estimating prevalence of these mental health conditions, understanding their relationship to gender-based violence (GBV), and assessing changes in response to an empowerment intervention.

### Methods

Mental health measures were added to the final data collection point of a two-year randomized controlled trial (RCT) evaluating an empowerment self-defense intervention. Statistical models evaluated how past sexual violence, access to money to pay for a needed hospital visit, alcohol use, and self-efficacy affect both mental health outcomes as well as how the intervention affected female students' mental health.

### Findings

Population prevalence of mental health conditions for combined male and female adolescents was estimated as: PTSD 12.2% (95% confidence interval 10.5–15.4), depression 9.2% (95% confidence interval 6.6–10.1) and anxiety 17.6% (95% confidence interval

data describing PTSD, anxiety, and depression, and (c) the age of the participants (ranging between 10 and 14 years at baseline). The data will therefore only be available via request directly to the corresponding author or a representative at Stanford University. The representative was not involved in the study and will vet that IRB and data safety guidelines are appropriately in place. The representative is Bonnie Halpern-Felsher, PhD, Professor of Pediatrics and (by courtesy) Health Research and Policy at Stanford University School of Medicine, and can be reached at bonniehalpernfelsher@stanford.edu. The corresponding author is Rina Friedberg, and she can be reached at rinafriedberg@gmail.com.

**Funding:** This study was funded by South African Medical Research Council through the What Works to Prevent Violence Innovation Grant (#52069); by the Department of Defense, Air Force Office of Scientific Research, National Defense Science and Engineering Graduate (NDSEG) Fellowship, 32 CFR 168a; and by the Marjorie Lozoff Fund, Michelle R. Clayman Institute for Gender Research, Stanford University. LinkedIn Corp provided support in the form of salary for RF. The funders had no role in study design, data collection and analysis, decision to publish, or preparation of the manuscript. The specific roles of these authors are articulated in the 'author contributions' section.

**Competing interests:** The authors have read the journal's policy and have the following competing interests: RF is a paid employee of LinkedIn Corp. This does not alter our adherence to PLOS ONE policies on sharing data and materials. There are no patents, products in development or marketed products associated with this research to declare. The other authors do not have any competing interests.

11.2% - 18.7%). Female students who reported rape before and during the study-period reported significantly higher incidence of all mental health outcomes than the study population. No significant differences in outcomes were found between female students in the intervention and standard-of-care (SOC) groups. Prior rape and low ability to pay for a needed hospital visit were associated with higher prevalence of mental health conditions. The female students whose log-PTSD scores were most lowered by the intervention (effects between -0.23 and -0.07) were characterized by high ability to pay for a hospital visit, low agreement with gender normative statements, larger homes, and lower academic self-efficacy.

## Conclusion

These data illustrate a need for research and interventions related to (1) mental health conditions among the young urban poor in low-income settings, and (2) sexual violence as a driver of poor mental health, leading to a myriad of negative long-term outcomes.

## Background

Gender-based violence (GBV) and mental health conditions are distinct but intertwined global challenges [1–4]. For example, in the Democratic Republic of the Congo, territories reporting high rates of sexual violence and human rights violations also reported higher rates of adult depression and post-traumatic stress disorder (PTSD) [5]. In the United States, adult women who had been raped also reported higher rates of PTSD and depression compared to those who had not been raped [6]. This pattern has also been documented for women living in shelters [7, 8], for a wide range of women across the United States [9], and for adolescents in global urban environments [10].

Adolescence is a critical time for prevention of both GBV and mental illness. Although sequelae can manifest later in life, the World Health Organization (WHO) estimates that half of all mental illnesses begin by age 14 and three-quarters by age 20. Neuropsychiatric conditions are the leading cause of disability, and suicide is the second-leading cause of death, among 15–29 year-olds globally [11]. Thus, approaches that might prevent or mitigate mental health problems in this age group are essential. Research into these approaches must take into account different experiences for severely underserved populations such as the one we study here.

In Nairobi, Kenya, adolescents living in informal urban settlements are exposed to high rates of poverty, violence, and other traumas, yet little is known about the impact of this environment on mental health. Rates of GBV in these settlements are high: for example, depending on age, between 8–25% of female adolescents living in the informal settlements experience sexual assault each year [12–14]. These numbers are much higher than the estimated nationwide incidence of 11% among females and 4% among males aged 13–17 [15]. Previously-described mental health consequences of GBV among adolescents in other settings include anxiety, PTSD, depression, and suicidality [16–19]. These consequences are exacerbated by high rates of poverty, as poor mental health and poverty have been shown to interact in a negative cycle in low- and middle-income countries [20]. Mental health challenges have also been linked to higher odds of cardiovascular, arthritis, respiratory, and chronic pain conditions, especially if the initial trigger was sexual violence [21].

In this study, we both consider mental health overall in this population and in response to an empowerment self-defense (ESD) program. ESD programs, which teach skills like awareness, verbal confrontations, and physical self-defense, have been shown to reduce annual incidence of sexual assault [12, 22, 23]. Moreover, several studies have found relationships between empowerment trainings and mental health and well-being outcomes. A self-defense and psychoeducation intervention reduced PTSD and depression among female veterans [24]; another was shown to reduce anxiety among women who attended a physical self-defense course [25]. Similar programs have also been shown to increase female participants' self-efficacy [26, 27], as well as reducing self-blame for past assaults and increasing ability to recognize abusive situations and think positively about gender [22, 27, 28].

As discussed above, informal settlements have high prevalence of sexual violence, which has been demonstrated to negatively affect mental health, and the global burden of poor adolescent mental health is enormous and poorly characterized. Therefore, we sought to understand the potential benefits of an ESD intervention on adolescent girl's mental health, as well as describe overall adolescent mental health, in these under-studied communities.

The primary aims of the current study were the following: (1) estimate the prevalence of depression, anxiety, and PTSD among adolescents living in the informal settlements of Nairobi, both overall and stratified by sex; (2) stratify prevalence of mental health conditions by history of sexual violence; (3) explore the potential average treatment effect of the empowerment intervention on mental health (i.e., the population level effect of the intervention on all participants); and (4) investigate possible sources of heterogeneity in those treatment effects.

## Ethics

Global ethical and safety guidelines for research on gender-based violence and with adolescents were followed. Data collectors were trained in trauma-informed interviewing practices and all had lists of resources for mental and physical health services if a young person disclosed a need for such services. All data collection was confidential, with surveys identified by unique ID numbers, and Stanford data analysts were blinded to which adolescents reported experiencing mental health symptoms. Data were collected onsite at the schools, recorded on paper, and then entered online for secure transmission. RedCap was used to enter and transfer data from the Nairobi site to Stanford to maintain security and privacy of data throughout the study. This is an add-on to a clinical trial originally registered via ClinicalTrials.gov, #NCT02771132. This study had IRB approval at Stanford University and from the Kenya Medical Research Institute (KEMRI). Written consent was obtained from parents, and assent was obtained from students. Participants could choose to opt out of any part of, or the entire, survey at any time.

## Methods

### Sample

The described study built on a larger cluster-randomized control trial (CRT), implemented for 4,091 class eight students (3,263 female and 828 male), that evaluated the effectiveness of an ESD intervention for reducing annual rape incidence among female students. Data collection for this RCT occurred between January 2016 and December 2018.

### Intervention

The female students' intervention involved empowerment, gender norms, techniques for achieving goals, and self-defense [29]. The male students' intervention focused on gender norms and achieving positive social goals. The intervention was deployed at the school level.

Each program involved 12 total hours of classroom-based sessions, and data was collected at baseline and after a follow-up period of about two years. The standard of care (SOC) group received training on life skills such as hygiene, citizenship, and financial planning.

The CRT took place in five informal settlements outside of Nairobi, denoted neighborhoods A-E for confidentiality. Schools were selected for their willingness to participate in the study, and a final total of 94 schools, with one class per school, received either the intervention or SOC. Baseline data was collected and analyzed after randomization. The follow-up data collection round for the larger CRT, conducted May through November of 2018, included the mental health scales. Fig 1 gives the CONSORT flow diagram for the CRT (female students).

## Measures

All mental health measures had been previously tested and validated in adolescent populations. Specifically, these included (1) the Child PTSD Symptom Scale (CPSS), (2) the Beck Child Depression Inventory 2 (CDI 2), (3) the Beck Anxiety Inventory (BAI), and (4) the Rosenberg self-efficacy scale (RSES). Many of these scales have been used globally, and have been piloted with sexual assault survivors specifically [30, 31].

History of sexual violence was measured by asking students a series of questions about experiences with sexual violence; for example, "In the past 12 months, how many times has a current or a previous boyfriend ever physically forced you to have sex when you did not want to?". Their responses were then aggregated with an adjudication model, previously described elsewhere [32, 33].

## Data analysis

Throughout this paper, we considered prevalence of PTSD, depression, and anxiety scored as moderate to severe; these correspond to cutoffs of 21, 20, and 17, for each respective mental health condition, consistent with previous literature [34–36]. In order to create a population estimate of these mental health outcomes, there were two main issues to balance.

The first was the male-female balance. There were 3,263 female students but only 828 male students at the data collect point. Given the dearth of data describing mental health for male and female adolescents in this population, we sought to derive population estimates targeting gender parity. To facilitate comparisons between female and male groups, we gave the total male students from each school equal weight to the total female students from that school. For all outcomes, we provide 95% confidence intervals based on 1000 clustered bootstrap replicates [37].

The second balance issue is that students who dropped-out between the beginning of the CRT and the endpoint, where this mental health data was collection (n = 858), did not do so at random, but rather those at higher risk of rape dropped out at higher rates [38]. To address this, we used inverse probability weights [39] to "upweight" students who remained in our study—but who appear similar to those who dropped out—in order to mitigate the effect of the dropout. All results discussed here are balanced by inverse probability weights. Other missing data were omitted from the analysis.

## Analysis objective 1: Mental health condition prevalence

Overall prevalence estimates for moderate-to-severe mental health conditions were calculated for the overall population and each gender by weighting the estimates for male and female students to estimate a 50/50 population balance. Furthermore, all were adjusted using inverse probability weights to address dropout.

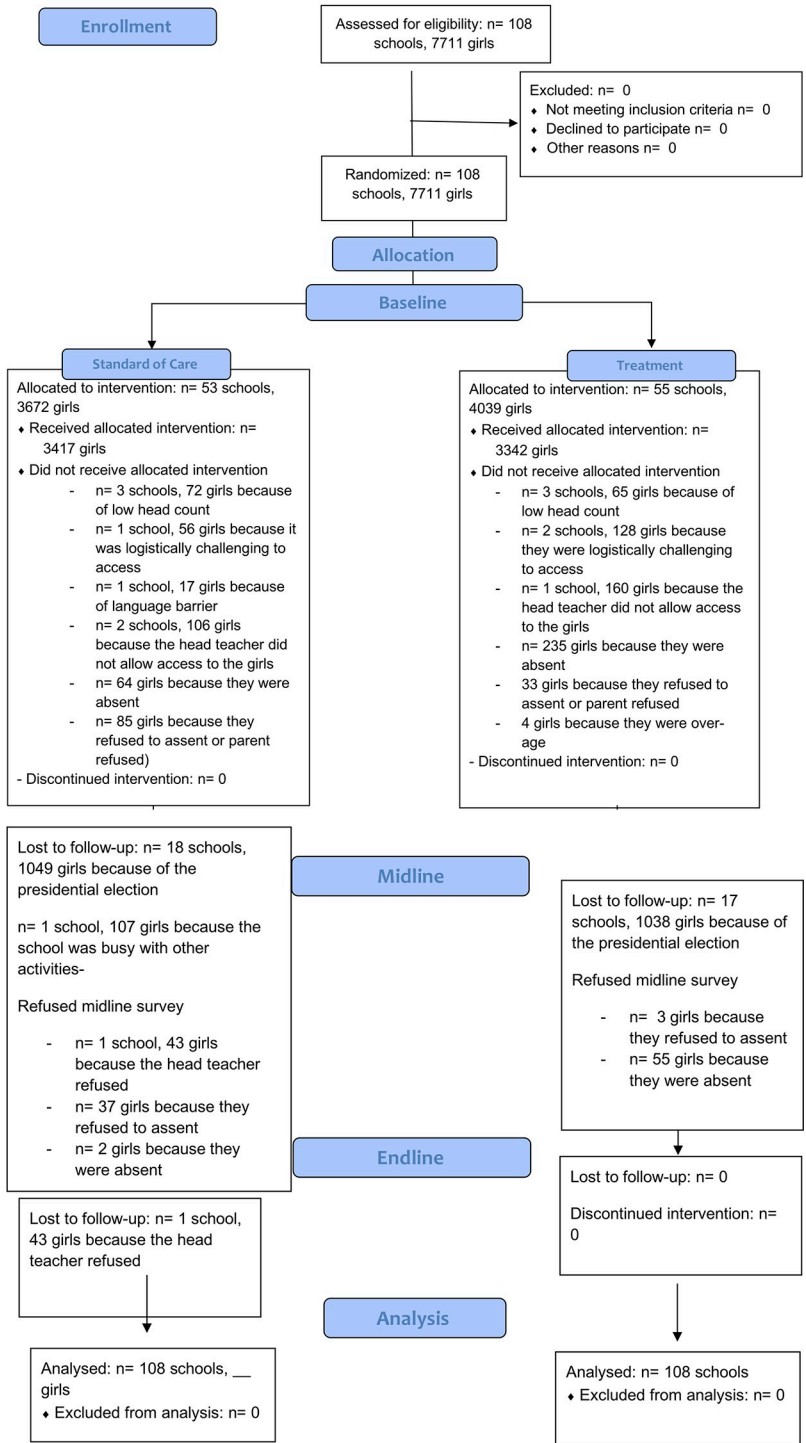

**Fig 1. CONSORT flow diagram.** Flowchart showing participants and schools from baseline to endline.

### Analysis objective 2: Mental health condition prevalence stratified by rape history

Descriptive analysis compared prevalence of mental health conditions between participants with a previous history of sexual assault, and those without. We also trained a linear mixed model to predict mental health conditions at the student level, using individual covariates including history of rape.

### Analysis objective 3: Understand the relationship between the empowerment intervention and mental health for female students

We trained a linear mixed effects model to predict mental health outcomes from intervention assignment, neighborhood, baseline rape incidence, home size, and a socioeconomic status measure asking if a student could obtain money for a needed hospital visit. Covariates used in the models to predict 2-year responses to the intervention were all collected from baseline measures and previously identified as relevant features (39). The random assignment mechanism tended to produce balanced assignments, and moreover we observed high-quality balance across many covariates at baseline [33], providing confidence in our point estimate of the treatment effect. This model assumed a random effect for each school; while outcomes may differ by school, we did not aim to model that difference. Outcomes were log-transformed to reduce the undue influence of outlying points. Coefficients are as reported in R by the package lme4 [40]. We report standard errors estimated by bootstrapping the full procedure 1000 times, and report significance at the $p = 0.05$ level if 0 is not contained in the 95% bootstrap confidence interval.

### Analysis objective 4: Investigate possible sources of heterogeneity in intervention responses

Last, we considered individual causal effects of the empowerment intervention on mental health measures. We predicted the conditional average treatment effect (CATE); here "conditional" indicates that this is the predicted treatment effect conditional on the covariates we observe about an individual. A local linear causal forest model for individual log PTSD scores was used. The model was trained with inverse probability of dropout weights. Unlike our prior analysis, this method assumes no specific model and instead adaptively learns patterns in the data [41].

This section focuses on PTSD for brevity; analogous results for depression and anxiety are included in S1 Fig. All scores are again scaled by the log transformation to manage outliers. Correspondingly, the prediction discussed here for one individual is the expected change in her log-transformed PTSD score if she received the empowerment intervention, compared to if she did not receive the intervention. Large, negative estimated values suggest the participant benefited quite a bit (i.e., a decrease in the mental health condition), whereas positive values suggest a worsening in response to the intervention.

## Results

### Objective 1: Overall prevalence

The two-year retention rate for female students was 79.2%, with 4,121 completing the survey at baseline and 3,263 of those completing the follow-up survey. For male students, retention was 75.3%, with a baseline total of 1,105 students and a follow-up total of 832 students.

Population prevalence of depression, anxiety, and PTSD for combined male and female class 8 students, as well as for female and male students separately, can be found in Table 1.

**Table 1. Prevalence of moderate to severe PTSD, depression, and anxiety.**

| Group | PTSD | 95% CI | Depression | 95% CI | Anxiety | 95% CI |
|---|---|---|---|---|---|---|
| Overall | 12.2 | (10.5, 15.4) | 9.2 | (6.6, 10.1) | 17.6 | (11.2, 18.7) |
| Female | 11.6 | (10.4, 12.9) | 10.4 | (9.1, 11.7) | 18.9 | (16.5, 21.4) |
| Female, $R_0 = 0$ | 10.7 | (9.4, 12.0) | 9.3 | (8.1, 10.6) | 17.6 | (15.2, 20.3) |
| Female, $R_0 = 1$ | 21.6 | (16.6, 26.7) | 21.9 | (17.2, 27.4) | 34.0 | (24.7, 43.8) |
| Male | 12.9 | (10.2, 15.5) | 5.9 | (4.3, 8.3) | 16.7 | (10.5, 21.4) |
| Male, $R_0 = 0$ | 13.2 | (10.3, 16.1) | 5.5 | (3.5, 8.2) | 15.9 | (9.9, 20.6) |
| Male, $R_0 = 1$ | 9.9 | (2.6, 17.3) | 10.7 | (3.7, 17.7) | 26.4 | (14.0, 41.9) |

The notation $R_0$ indicates whether a student reported rape at the beginning of the study, and $R_1$ indicates whether they reported rape at follow-up.

Overall population estimates were: PTSD 12.2% (95% confidence interval 10.5%, 15.4%), depression 9.2% (6.6%, 10.1%) and anxiety 17.6% (11.2%, 18.7%). By gender, female students reported significantly higher levels of depression (10.4%; 95% CI 9.1%-11.7%) compared to male students (5.9%, 95% CI 4.3%-8.3%). The other comparisons across gender were not statistically significant.

## Objective 2: Prevalence by history of sexual violence

Table 1 additionally gives prevalence statistics among class 8 female and male students, broken into students without ($R_0 = 0$) and with ($R_0 = 1$) reported prior rape at the baseline data collection period. Female students who had not been raped before the baseline study period reported overall PTSD prevalence of 10.7% (95% CI 9.4, 12.0), while those who had been raped before the study reported a prevalence of 21.6% (16.6, 26.7). Female students who reported rape before the baseline study period also reported significantly higher depression (21.9%, 17.2–27.4, compared to 9.3%) and anxiety (34.0%, 24.7–43.8, compared to 17.6%). Moreover, among female adolescents who had not been raped at baseline, but reported a rape during the study period, rates of all three outcomes were high, with PTSD at 38.7% (intervention) comped to 33.1% (SOC); depression at 31.0% (intervention) and 22.6% (SOC); and anxiety at 44.8% (intervention) and 44.5% (SOC). Fig 2 shows these results graphically, with the estimated overall prevalence shown for comparison. Estimates are displayed as the black squares, along with

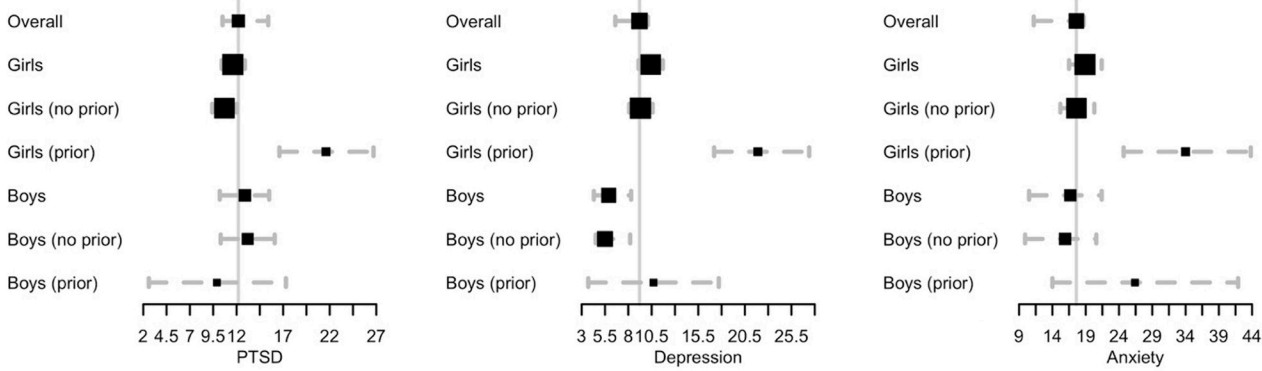

**Fig 2. Prevalence of mental health conditions.** Forest plots of prevalence estimates for moderate-to-severe PTSD, depression, and anxiety.

**Table 2. Results of separate mixed effects Gaussian models predicting log of each individual PTSD, depression, and anxiety for female adolescents.**

| Fixed effect | Coefficient (PTSD) | | Coefficient (depression) | | Coefficient (anxiety) | |
|---|---|---|---|---|---|---|
| | Estimate | Standard error | Estimate | Standard error | Estimate | Standard error |
| Intercept | 1·85* | 0·07 | 1·98* | 0·08 | 2·07* | 0·10 |
| Prior rape at baseline | 0·35* | 0·08 | 0·24* | 0·07 | 0·18* | 0·10 |
| Ability to obtain money for a needed hospital visit | -0·03* | 0·02 | -0·05* | 0·01 | -0·06* | 0·02 |
| Home size | 0·01 | 0·01 | 0·00 | 0·01 | -0·01 | 0·01 |
| Area- A | -0·26* | 0·08 | 0·06 | 0·07 | 0·53* | 0·11 |
| Area- B | -0·32* | 0·08 | -0·07 | 0·07 | 0·24* | 0·09 |
| Area- C | -0·41* | 0·06 | -0·17* | 0·06 | -0·09 | 0·11 |
| Area—D | -0·02 | 0·06 | 0·11 | 0·06 | 0·26* | 0·08 |
| Treatment | 0.02 | 0.04 | -0.03 | 0.03 | 0.05 | 0.06 |

*significant at 0.05 level

95% confidence intervals shown as dotted gray lines. The central gray line is the overall estimated prevalence.

We also analyzed covariates associated with the mental health outcomes (Table 2) at follow-up, aggregating across arms of the intervention. Those that were significant across all three categories (PTSD, depression, and anxiety) were: (1) prior rape at baseline (coefficients of 0.35, 0.24, and 0.18, respectively), which increased negative mental health condition scores, and (2) the ability to obtain money for a needed hospital visit (coefficients -0.03, -0.05, and -0.06, respectively), which decreased negative mental health condition scores.

## Objective 3: Effect of the empowerment intervention

To quantify the effect of the ESD intervention on mental health, we compared the prevalence of mental health outcomes between intervention and SOC populations (Table 3), as well as between the sub-group of individuals from intervention and SOC populations who reported rape. Though not significant, the strongest relationship detected in the exploratory analysis was the differential rate of depression between the SOC group (11.6%, 95% CI 9.9–13.3) compared to the intervention group (9.2%).

## Objective 4: Intervention effect heterogeneity

Fig 3 shows a histogram of the resulting estimated individual treatment effects; with a mix of negative and positive CATEs. The left panel shows a plot of CATE estimates, and the right plot

**Table 3. Prevalence of PTSD, depression, and anxiety by treatment status.**

| Group | PTSD | 95% CI PTSD | Depression | 95% CI depression | Anxiety | 95% CI anxiety |
|---|---|---|---|---|---|---|
| Treatment | 11.0 | (9.3, 12.9) | 9.2 | (7.4, 11.0) | 18.5 | (15.2, 21.9) |
| SOC | 12.3 | (10.4, 14.4) | 11.6 | (9.7, 13.6) | 19.4 | (15.9, 23.2) |
| Treatment with $R_1 = 0$ | 10.2 | (8.3, 11.8) | 8.6 | (7.0, 10.3) | 17.7 | (14.4, 21.4) |
| SOC with $R_1 = 0$ | 11.3 | (9.3, 13.3) | 10.0 | (8.1, 11.9) | 17.5 | (14.0, 20.9) |
| Treatment with $R_0 = 0$, $R_1 = 1$ | 38.7 | (28.8, 49.6) | 31.0 | (23.2, 40.2) | 44.8 | (25.2, 62.0) |
| SOC with $R_0 = 0$, $R_1 = 1$ | 33.1 | (23.1, 44.1) | 22.6 | (13.3, 33.9) | 44.5 | (26.7, 66.7) |

Estimates in Table 3 are weighted by estimated dropout probability.

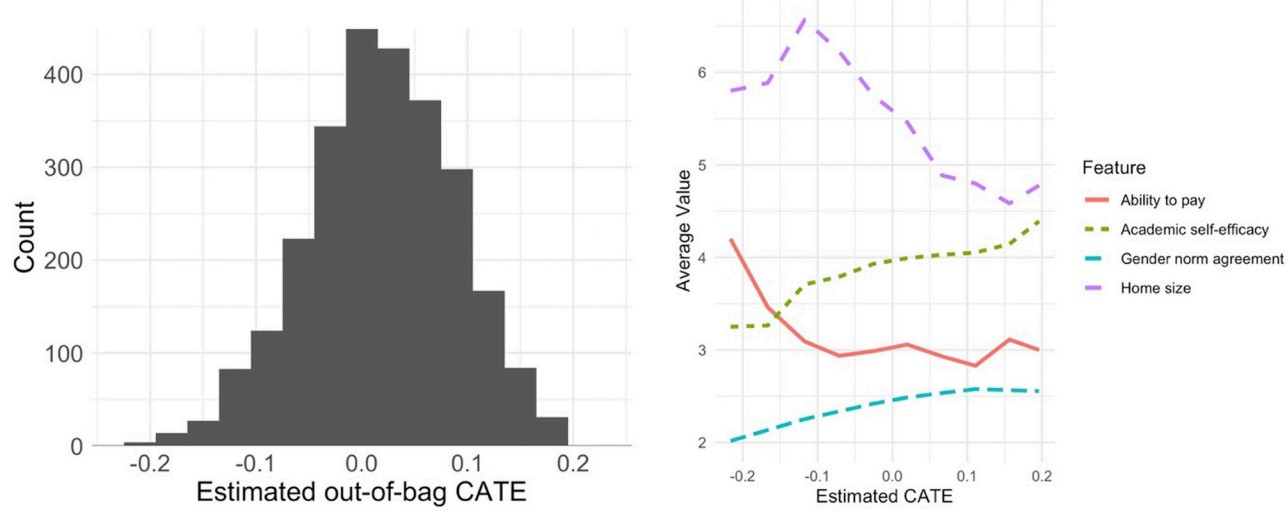

**Fig 3. Conditional average treatment effects.** CATE estimates, shown as a histogram (left) and varying with covariate values (right).

shows average covariate values that were significantly different between low-CATE individuals and the other two groups, averaged by 10 quantiles of predicted CATE. The prediction shown in the left panel, for one individual, is the expected change in her log-transformed PTSD score if she received the empowerment intervention, compared to if she did not receive the intervention. Large, negative estimated values suggest the participant benefited quite a bit (i.e., a decrease in PTSD), whereas positive values suggest a worsening in response to the intervention. The right panel also shows that the female adolescents whose log-PTSD scores were most lowered by the intervention (effects between -0.23 and -0.07) were characterized by high ability to pay for a hospital visit, low agreement with negative gender normative statements, larger homes, and lower academic self-efficacy.

Table 4 gives average covariate values for female students with low (indicating lower PTSD due to the intervention), middle (close to zero, indicating no effect), and high estimated CATEs. Covariates included are average agreement with gender-normative statements such as "a woman wearing a short skirt is 'asking for it'"; academic, social, and emotional self-efficacy, defined as an individual's belief in her capabilities; the number of individuals in a home; ability

**Table 4. Average baseline covariate values, with 95% bootstrap confidence intervals, for three regions of the conditional average treatment effect (CATE).**

| Variable | Low-CATE | 95% CI | Mid-CATE | 95% CI | High-CATE | 95% CI |
|---|---|---|---|---|---|---|
| Gender norm agreement | 2.23 | (2.18, 2.27) | 2.46 | (2.45, 2.48) | 2.57 | (2.55, 2.58) |
| Academic self-efficacy | 3.74 | (3.65, 3.83) | 3.95 | (3.92, 3.97) | 4.09 | (4.04, 4.15) |
| Home size | 6.37 | (6.06, 6.67) | 5.51 | (5.39, 5.63) | 4.68 | (4.67, 4.89) |
| Social self-efficacy | 3.72 | (3.62, 3.82) | 3.90 | (3.87, 3.92) | 3.56 | (3.51, 3.61) |
| Emotional self-efficacy | 3.74 | (3.63, 3.84) | 3.76 | (3.73, 3.79) | 3.29 | (3.23, 3.35) |
| Ability to pay for a hospital visit | 3.15 | (2.99, 3.16) | 2.98 | (2.93, 3.04) | 2.93 | (2.82, 3.05) |
| Frequency of monthly alcohol use | 1.03 | (0.17, 2.38) | 2.04 | (1.49, 2.66) | 1.57 | (0.61, 2.49) |
| Violence of fathers against mothers | 0.15 | (0.08, 0.21) | 0.19 | (0.16, 0.21) | 0.19 | (0.15, 0.24) |
| Prior rape at baseline | 0.12 | (0.08, 0.17) | 0.09 | (0.08, 0.11) | 0.05 | (0.04, 0.08) |

to pay for a needed hospital visit; frequency of alcohol use per month at baseline; history of violence from a student's father against her mother; and prior rape at study baseline. We report average unweighted covariate values for the three groups of students with low (-0.23 to -0.07), mid (-0.07 to 0.08), and high (0.08 to 0.24) estimated CATEs. Low-CATE individuals were characterized by significantly lower agreement with gender-normative statements (2.23; 95% CI 2.18–2.27; mid-CATE 2.46, high-CATE 2.57) and greater ability to pay for a needed hospital visit, compared to both other groups (3.15, 95% CI 2.99–3.16; mid-CATE 2.98, high-CATE 2.93).

## Discussion

Mental health conditions among socioeconomically disadvantaged adolescents, particularly those in low-income settings, are critically understudied. A major contribution of our work is a set of estimates for the prevalence of moderate to severe PTSD (12.2%), depression (9.2%) and anxiety (17.6%) among male and female class eight students attending schools in informal settlements of Nairobi, Kenya.

The literature offers some insights into semi-comparable populations. For example, our study found a substantially higher rate of PTSD relative to a reported rate (6.6%) for children aged 3–19 in rural Uganda [42]. In contrast, our depression and anxiety rates were similar to prior findings of female adolescents in several countries, ranging from 11.2% in Sudan [43] to 15.3% in Cairo, Egypt [44]. The wide range of anxiety prevalence we found is also consistent with the literature. For example, reported adolescent anxiety prevalence estimates have ranged from 15% in a Nigerian population [45] to 26.6% in a Ugandan population [42].

In this sample, depression varied notably by gender, with female adolescents reporting significantly higher prevalence, at 10.4%, than male adolescent, at 5.9%. This finding is consistent with many other studies reporting higher depression rates among women and adolescent girls globally [46, 47]. Similar variation by gender did not, however, hold for anxiety or PTSD, which was an unexpected finding based on the literature. Future work should consider whether this pattern would hold for PTSD and anxiety in a larger study of this population.

We also found that experiencing rape within the last two years was a strong predictor of scoring poorly on all PTSD, depression, and anxiety scales. This relationship held regardless of being in the intervention or control group of the larger study. It is also consistent with prior research that has established that experiencing rape during adolescence is correlated with depression, anxiety, and PTSD [46, 48, 49].

We found that reporting prior rape at baseline was especially detrimental to the mental health of these young adolescents. Specifically, our generalized linear model associated prior rape at baseline with significantly increased rates of PTSD, depression, and anxiety. That model also suggested that the ability to pay for a needed hospital visit for a family member was a significantly protective factor for PTSD and depression. These results indicate that students with more favorable baseline conditions, both in terms of rape experiences and economically, are less likely to report mental health conditions. It also illustrates a dire need for mental health services that are accessible to the most vulnerable populations who may be most at risk for mental health conditions.

Ability to pay for a needed hospital visit for a family member also showed up as a factor associated with individuals who had the largest mental health improvements over the duration of the study. This group also had significantly less agreement with negative gender-normative statements as well as significantly lower academic self-efficacy; the latter merits future exploration. Combining these results with the generalized linear mixed model that showed that ability to pay for a needed hospital visit was also protective for mental health conditions, there is a cohesive story, consistent with other literature, about how individuals who start off in more stable positions are more likely to remain in stable positions.

Finally, we compared differences between the cohorts of female adolescents who received the empowerment intervention versus the SOC, but found no statistically significant differences in mental health outcomes. As the empowerment intervention was designed for GBV reduction, not targeted to mental health outcomes, this is expected. Nonetheless, since poorer mental health and experiences of violence are often correlated, it may be worth including mental health measures in GBV evaluations, and consider combined interventions that might be able to address both of these intertwined issues concurrently [50].

## Limitations

The major limitation of this study is that it is primarily exploratory in nature; the intervention and parent RCT was designed to study rape, not mental health. Replications of this research, with baseline measurements of mental health, and analysis of empowerment interventions focused on mental health, are warranted. In addition, as these questions were placed at the end of the survey, response rates were lower than ideal. For example, the anxiety questions were only answered by about half of the respondents, and it is likely that students unable to complete the survey in the allotted time were non-randomly different than those who were. Nonetheless, these methods are applicable to other populations and contexts, and this study demonstrates how rigorous statistical methods can be deployed to analyze global mental health.

## Diversity

This work describes the mental health of adolescents living in the informal settlements of Nairobi, who constitute a diverse population compared to those traditionally studied in academic journals. There is limited data and research describing mental health and sexual violence in such low-income populations, especially among this age group. This study adds to the literature connecting gender-based violence to mental health, supporting the evidence base that sexual violence is an important driver of poor mental health for adolescents, and broadening the knowledge base to include adolescents living in informal urban settlements.

## Summary

We observed high prevalence of mental health conditions, and large differences corresponding to experiences of sexual violence, in adolescents living in informal settlements in this study. These findings suggest a need for more research and programming with a focus on the relationship between mental health and sexual violence in these settings. This study also has implications for policy makers and funders. In particular, the findings that young people who are raped as fairly young children, as well as those with larger economic disadvantages, are more likely to report mental health challenges speaks to unmet need. Few prevention interventions for either sexual assault or poor mental health outcomes are aimed at young people before their teenage years; but it seems possible that such targeting to younger populations might help mitigate health issues and related expenses that have the potential to be life-long. Furthermore, understanding the causal relationship between sexual assault and mental health conditions, and the reverse, may allow more interventions to tackle both challenges simultaneously in order to improve the health of these poorly served populations.

## Supporting information

**S1 Fig. Histograms of predicted CATE values for the female students.** We consider depression (left) and anxiety (right).
(TIFF)

**S1 Table. Prevalence of mental health disorders, stratified by overall rates, sex, and prior violence experience.** These estimates are not weighted by probability of dropout during the study.
(DOCX)

**S2 Table. Heterogeneity checks across treatment vs. control, unweighted.** Prevalence and 95% bootstrap confidence intervals are displayed for each mental health outcome and each subgroup, without IPW.
(DOCX)

## Author Contributions

**Conceptualization:** Michael Baiocchi, Mary Amuyunzu-Nyamongo, Clea Sarnquist.

**Data curation:** Rina Friedberg, Michael Baiocchi, Mary Amuyunzu-Nyamongo, Gavin Nyairo.

**Formal analysis:** Rina Friedberg, Evan Rosenman.

**Funding acquisition:** Michael Baiocchi, Clea Sarnquist.

**Methodology:** Rina Friedberg, Michael Baiocchi, Evan Rosenman, Clea Sarnquist.

**Project administration:** Mary Amuyunzu-Nyamongo, Gavin Nyairo.

**Resources:** Michael Baiocchi, Mary Amuyunzu-Nyamongo, Gavin Nyairo, Clea Sarnquist.

**Software:** Rina Friedberg.

**Validation:** Rina Friedberg, Evan Rosenman.

**Visualization:** Rina Friedberg.

**Writing – original draft:** Rina Friedberg, Michael Baiocchi, Clea Sarnquist.

**Writing – review & editing:** Rina Friedberg, Michael Baiocchi, Mary Amuyunzu-Nyamongo, Gavin Nyairo, Clea Sarnquist.

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
