## [Decision Letter · Decision Letter 0]

18 May 2022

PONE-D-21-25156Mental health and gender-based violence: An exploration of depression, PTSD, and anxiety among adolescents in informal settlements participating in an empowerment interventionPLOS ONE

Dear Dr. Friedberg,

Thank you for submitting your manuscript to PLOS ONE; I sincerely apologise for the unusually delayed review timeframe. After careful consideration, we feel that it has merit but does not fully meet PLOS ONE’s publication criteria as it currently stands. Therefore, we invite you to submit a revised version of the manuscript that addresses the points raised during the review process.

Please note that we have only been able to secure a single reviewer with expertise in biostatistics to assess your manuscript. We are issuing a decision on your manuscript at this point to prevent further delays in the evaluation of your manuscript. Please be aware that the editor who handles your revised manuscript might find it necessary to invite additional reviewers to assess this work once the revised manuscript is submitted. However, we will aim to proceed on the basis of this single review if possible.

We look forward to receiving your revised manuscript.

Kind regards,

Emily Chenette

Editor in Chief

PLOS ONE

**Journal requirements:**

5.  Please upload a copy of Figure 3, to which you refer in your text on page 15. If the figure is no longer to be included as part of the submission please remove all reference to it within the text.

Reviewers' comments:

Reviewer's Responses to Questions

**Comments to the Author**

1. Is the manuscript technically sound, and do the data support the conclusions?

Reviewer #1: Partly

2. Has the statistical analysis been performed appropriately and rigorously? 

Reviewer #1: No

3. Have the authors made all data underlying the findings in their manuscript fully available?

Reviewer #1: Yes

4. Is the manuscript presented in an intelligible fashion and written in standard English?

Reviewer #1: Yes

5. Review Comments to the Author

Reviewer #1: Abstract methods: need details for the study design and statistical methods.

The statistical analysis is not clearly written.

Objective 1:

For table 2, clarify whether the outcomes are baseline or change. The mixed model was not mentioned in method section of Objective 1 (mentioned only in the title). Why separate female and male? Gender can be added as a covariate and its interactions with other covariates can be evaluated. Was log of score used? Normality tested?

More important of all, better use logistic regression to model prevalence of PTSD, depression and anxiety as they are the primary outcomes for this objective.

Further, Table 2 does not seem to belong here as the objective is for mental health prevalence, not associations with baseline characteristics.

Objective 2:

What model did you use for GBV comparisons? Better use logistic regression rather than “Descriptive analysis”. GBV can be evaluated in the model. Need odds ratios and p values.

Objective 3:

The “mixed” model mentioned in the method section does not match the results presented in Table 3. The method section mentioned “outcomes were log-transformed” but the outcomes in Table 3 are all dichotomous.

To evaluate the intervention effects, the analysis should focus on the comparisons between intervention and control. Logistic regression is more appropriate for dichotomous outcomes. GBV with other characteristics can be added as covariates.

Objective 4:

How are low, mid and high-CATE determined? Are those cutoff points clinically meaningful?

Predictors in Table 4 need to be clearly described.

Need rationale to use log-transformation of outcomes. Does it improve normality?

Why bootstrap? As this is a large study.

Flow charts: clarify N is # of schools.

6. PLOS authors have the option to publish the peer review history of their article (what does this mean?). If published, this will include your full peer review and any attached files.

Reviewer #1: No

---

## [Author Response · Author response to Decision Letter 0]

3 Aug 2022

We thank the editor and the reviewer for their helpful comments! Responses to each individual point are included in the response to reviewers.

---

## [Decision Letter · Decision Letter 1]

31 Oct 2022

PONE-D-21-25156R1Mental health and gender-based violence: An exploration of depression, PTSD, and anxiety among adolescents in informal settlements participating in an empowerment intervention

PLOS ONE

Dear Dr. Friedberg,

Thank you for submitting your manuscript to PLOS ONE. After careful consideration, we feel that it has merit but does not fully meet PLOS ONE’s publication criteria as it currently stands. Therefore, we invite you to submit a revised version of the manuscript that addresses the points raised during the review process.

Your revised manuscript has been reviewed by the original peer-reviewer and additional reviewers and their reports are appended below. 

The reviewers comment that your manuscript would benefit from additional details on the study design, methodology and data analysis. In addition, the reviewers request that terms used in the manuscript are better defined or further clarified. Furthermore, the reviewers comment that the discussion section could benefit from further discussion regarding the implications and recommendations of the findings of this study on current and future studies and policies.

Could you please carefully revise the manuscript to address all comments raised?

We look forward to receiving your revised manuscript.

Kind regards,

Maria Elisabeth Johanna Zalm, Ph.D

Editorial Office

PLOS ONE

Reviewers' comments:

Reviewer's Responses to Questions

**Comments to the Author**

1. If the authors have adequately addressed your comments raised in a previous round of review and you feel that this manuscript is now acceptable for publication, you may indicate that here to bypass the “Comments to the Author” section, enter your conflict of interest statement in the “Confidential to Editor” section, and submit your "Accept" recommendation.

Reviewer #1: All comments have been addressed

Reviewer #2: (No Response)

Reviewer #3: (No Response)

Reviewer #4: (No Response)

2. Is the manuscript technically sound, and do the data support the conclusions?

Reviewer #1: (No Response)

Reviewer #2: Yes

Reviewer #3: Yes

Reviewer #4: No

3. Has the statistical analysis been performed appropriately and rigorously? 

Reviewer #1: (No Response)

Reviewer #2: Yes

Reviewer #3: Yes

Reviewer #4: I Don't Know

4. Have the authors made all data underlying the findings in their manuscript fully available?

Reviewer #1: (No Response)

Reviewer #2: No

Reviewer #3: (No Response)

Reviewer #4: No

5. Is the manuscript presented in an intelligible fashion and written in standard English?

Reviewer #1: (No Response)

Reviewer #2: Yes

Reviewer #3: Yes

Reviewer #4: No

6. Review Comments to the Author

Reviewer #1: (No Response)

Reviewer #2: Thank you for the opportunity to review this paper. The current study aimed to explore the prevalence of mental health problems in terms of depression, anxiety and PTSD among adolescents in Kenya, Nairobi. Additionally, the study aimed to explore the relationship between these mental health problems and gender-based violence (GBV) as well as to explore changes in mental health outcomes in response to an empowerment intervention. Main findings were that the prevalence of mental health problems were high and more common in girls who had experienced GBV than those who had not. There were no differences in mental health outcomes between girls who had received the empowerment intervention and those who had not.

In my opinion, this is a robust and important study that adds knowledge to the current research field. Overall, the paper is well written, the statistical analyses are appropriate and the findings have clear implications for ways to improve the health for adolescents living in informal urban settlements. I also believe that the authors have responded well to the previous review comments. I do have a couple of minor comments/suggestions for how to improve the paper:

*Minor comment, at the end of the background section, line 93, “these communities”, which communities are you referring to?

*Currently, the study aims are described in the method section. I suggest moving them to the end of the background section in order to improve the flow of the text.

*In the method section, there is now description of the flow of participants/schools, e.g. number of dropouts. I understand that there was previously a flowchart describing this? I suggest to put it back.

*In the method section, please describe how many classes were included in the study.

*Were the schools randomized based on a 1:1 allocation ratio? Was the baseline assessment conducted prior to or after randomization? Please specify.

*In the method section, where you describe the mental health measures, please describe how GBV was assessed? Was this assessed with a scale or single items? Can you provide an example of such items? It is now briefly described in the analysis objective 2 section but I suggest to also add it here in order to improve the flow of the text.

*In the method section, more information could be provided on where the data collection took place. At the same location as where the CRT took place? I assume that the adolescents completed the measures on their mobile phones/ computers? How long approximately did the data collection take?

*In Tables 1 and 3, merely % of participants is provided. Is it possible to add N and total N to these tables? It would make them more informative.

Reviewer #3: Review of:

Mental health and gender-based violence: An exploration of depression, PTSD, and anxiety among adolescents in Kenyan informal settlements participating in an empowerment intervention.

Thank you for the opportunity to review this manuscript. It is believed ongoing research in the field of young people and specifically those living in circumstances that expose them to higher incidences of sexual trauma is much needed.

I have included a manuscript with track changes to refer to my specific comments. I include only my general comments below:

A strength of the manuscript is the statistical analyses. There are however some assumptions or decisions made about the data that requires more clarification in the manuscript.

The manuscript will benefit from the inclusion of a diagram to help depict the timelines of the research and the assessment points. As the research has multiple components the manuscript lacks some flow in my opinion. I have tried to comment on this in the attached track changes documents.

Reviewer #4: This study investigates are very under-researched sample in a low-income country which is a major strength of this work. The editor and reviewer of the first revision provided many helpful suggestions. The authors could add some more explanations in the response letter to the paper as readers wonder about the same things.

This is very important work and an interesting study design, but the manuscript needs much more work.

Some comments and suggestions:

Abstract:

Please state confidence intervals correctly (see CONSORT)

Please state statistical values in “findings”, at least p-values

First sentence in conclusion can already be read in findings

Background

Please define and explain ESDs

The background section is very short and really different paragraphs that don’t “make a story”

The significant gap in the literature is not well described

Methods

Please cite the RCT when first mentioning it (registration number, main paper, ….)

Introduce abbreviations only once

Aims of the current study need to be described in the background/ introduction section. Please describe more literature to each goal and state hypothesis

Please insert the common headings (sample, intervention,…)

Please describe measures more detailed with cut-off values

Please add the RCT flow-chart to better understand the high and systematic drop-out (I think this had been taken out afterwards? There is a crossed-out sentence on page 10)

Please describe handling of missings

I don’t understand the new sentence on page 8

Why was school not included as a control variable? The analysis could possibly also be implemented in a hierarchical structure

Results

I am not an expert in statistics, which is why I leave out comments on the analysis

Discussion

Please discuss the finding that female participants “only” report higher depression but no PTSD and anxiety, as this is not in line with the literature (p.18 top)

In the discussion section it is again not a coherent narrative but only sentences and paragraphs on a finding each

The conclusion on potential combined interventions (p.18-19) is not based on the results, please re-phrase

The discussion could benefit from more clinical and political implications of the results

7. PLOS authors have the option to publish the peer review history of their article (what does this mean?). If published, this will include your full peer review and any attached files.

Reviewer #1: No

Reviewer #2: No

Reviewer #3: No

Reviewer #4: No

---

## [Author Response · Author response to Decision Letter 1]

13 Dec 2022

Response to reviewers is included in the materials. Thank you very much for the feedback!

---

## [Decision Letter · Decision Letter 2]

2 Feb 2023

Mental health and gender-based violence: An exploration of depression, PTSD, and anxiety among adolescents in Kenyan informal settlements participating in an empowerment intervention

PONE-D-21-25156R2

Dear Dr. Friedberg,

We’re pleased to inform you that your manuscript has been judged scientifically suitable for publication and will be formally accepted for publication once it meets all outstanding technical requirements.

Kind regards,

Yann Benetreau

Staff Editor

PLOS ONE

Additional Editor Comments (optional):

* Please consider the requests by reviewers to address typos.

* Please list commercial affiliations as a competing interest; more information on our policy on competing interests is available at https://journals.plos.org/plosone/s/competing-interests

Reviewers' comments:

Reviewer's Responses to Questions

**Comments to the Author**

1. If the authors have adequately addressed your comments raised in a previous round of review and you feel that this manuscript is now acceptable for publication, you may indicate that here to bypass the “Comments to the Author” section, enter your conflict of interest statement in the “Confidential to Editor” section, and submit your "Accept" recommendation.

Reviewer #1: All comments have been addressed

Reviewer #2: All comments have been addressed

Reviewer #3: All comments have been addressed

2. Is the manuscript technically sound, and do the data support the conclusions?

Reviewer #1: (No Response)

Reviewer #2: Yes

Reviewer #3: Yes

3. Has the statistical analysis been performed appropriately and rigorously? 

Reviewer #1: (No Response)

Reviewer #2: Yes

Reviewer #3: Yes

4. Have the authors made all data underlying the findings in their manuscript fully available?

Reviewer #1: (No Response)

Reviewer #2: No

Reviewer #3: Yes

5. Is the manuscript presented in an intelligible fashion and written in standard English?

Reviewer #1: (No Response)

Reviewer #2: Yes

Reviewer #3: Yes

6. Review Comments to the Author

Reviewer #1: (No Response)

Reviewer #2: The authors have responded well to my previous comments and I believe the manuscript is suitable for publication.

Reviewer #3: Thank you for addressing all my concerns.

I have nothing further to add than small errors.

In the track changes document:

line 58: PTSD change to (PTSD)

line 390: which was ana unexpected finding

line 436: may be worth

are small typos

7. PLOS authors have the option to publish the peer review history of their article (what does this mean?). If published, this will include your full peer review and any attached files.

Reviewer #1: No

Reviewer #2: No

Reviewer #3: No

---

## [Editor Report · Acceptance letter]

17 Mar 2023

PONE-D-21-25156R2 

Mental health and gender-based violence: An exploration of depression, PTSD, and anxiety among adolescents in Kenyan informal settlements participating in an empowerment intervention 

Dear Dr. Friedberg:

I'm pleased to inform you that your manuscript has been deemed suitable for publication in PLOS ONE. Congratulations! Your manuscript is now with our production department. 

Kind regards, 

on behalf of

Dr. Maria Elisabeth Johanna Zalm 

Staff Editor

PLOS ONE